# An All-Silicon Resonant Pressure Microsensor Based on Eutectic Bonding

**DOI:** 10.3390/mi14020441

**Published:** 2023-02-13

**Authors:** Siyuan Chen, Jiaxin Qin, Yulan Lu, Bo Xie, Junbo Wang, Deyong Chen, Jian Chen

**Affiliations:** 1State Key Laboratory of Transducer Technology, Aerospace Information Research Institute, Chinese Academy of Sciences, Beijing 100190, China; 2School of Electronic, Electrical and Communication Engineering, University of Chinese Academy of Sciences, Beijing 100049, China

**Keywords:** resonant pressure sensor, eutectic bonding, thermal mismatch, temperature-insensitive, high accuracy

## Abstract

In this paper, an all-Si resonant pressure microsensor based on eutectic bonding was developed, which can eliminate thermal expansion coefficient mismatches and residual thermal stresses during the bonding process. More specifically, the resonant pressure microsensor included an SOI wafer with a pressure-sensitive film embedded with resonators, which was eutectically bonded with a silicon cap for vacuum encapsulation. The all-Si resonant pressure microsensor was carefully designed and simulated numerically, where the use of the silicon cap was shown to effectively address temperature disturbances of the microsensor. The microsensor was then fabricated based on MEMS processes where eutectic bonding was adopted to link the SOI wafer and the silicon cap. The characterization results showed that the temperature disturbances of the resonant pressure microsensor encapsulated with the silicon cap were quantified as −0.82 Hz/°C of the central resonator and −2.36 Hz/°C of the side resonator within a temperature range from −40 °C to 80 °C, which were at least eight times lower than that of the microsensor encapsulated with the glass cap. Compared with the microsensor using the glass cap, the all-silicon microsensor demonstrated an accuracy improvement from 0.03% FS to 0.01% FS and a reduction in short-term frequency fluctuations from 3.2 Hz to 1.5 Hz.

## 1. Introduction

Pressure microsensors based on MEMS have advantages of small sizes, high accuracies, low power consumptions, and low costs [1,2], and have been widely used in aerospace, industrial controls, weather monitoring, automotive electronics, and other fields [3,4,5]. According to different sensing principles, mature MEMS pressure microsensors can be mainly divided into piezoresistive pressure microsensors, capacitive pressure microsensors, and resonant pressure microsensors.

Piezoresistive pressure microsensors [6,7], which are mainly composed of pressure sensitive films and pieozresistors, use the piezoresistive effect of sensing materials to achieve pressure detections. Piezoresistive pressure microsensors have features including simple fabrication processes, high pressure sensitivities, small sizes, simple interface circuits, and low costs. However, the piezoresistive effect of silicon is strongly affected by temperature, thus piezoresistive pressure microsensors generally suffer from zero-point and thermal drifts [8,9].

Capacitive pressure microsensors [10,11,12] mainly consist of a movable electrode plate and a fixed electrode plate, in which the movable electrode plate lies on a pressure-sensitive film. They measure pressures by detecting changes in capacitance, which is affected by the distance between two electrode plates. Compared with piezoresistive pressure microsensors, capacitive pressure microsensors have advantages of high pressure sensitivities, low power consumptions, and fast responses [10]. Meanwhile, they have shortcomings of poor linearity, large parasitic capacitances, weak output signals, and low signal-to-noise ratios [11].

Resonant pressure microsensors [13,14,15] use the resonance principle to measure pressure variations, using resonators as sensitive elements to convert pressures to be measured into changes in resonant frequencies. Outputs of resonant pressure microsensors are resonant frequencies, which avoid accuracy degradation due to AD conversions and improve the anti-interference capabilities of the microsensors. Compared with piezoresistive and capacitive pressure microsensors, resonant counterparts have features including high stabilities, high repeatability, high resolutions, and high accuracies [16,17].

In 2008, Ma Z [18] et al. reported a resonant pressure microsensor based on SOI and electrostatic driving/capacitive detection. In this microsensor, a three-layer mask etching process was developed in which a silicon nitride mask was firstly used to pattern the device layer, followed by dry etching of the silicon oxide layer and a release by a TMAH solution. In the end, the microsensor was anodically bonded with a silica glass without vacuum encapsulation, thus the reported resonant pressure microsensor suffered from a low quality factor of 1200 in air and a low accuracy of 0.18% FS.

To achieve vacuum encapsulation, Du [19] et al. developed a resonant pressure microsensor in which a silicon-based resonator was vacuum encapsulated by a glass cap. The vacuum encapsulation improved the accuracy to 0.021% FS. However, owing to the mismatch between the coefficients of thermal expansions of resonators and vacuum caps, thermal stresses were generated on the resonators when the temperature changed, resulting in a temperature disturbance of 6.6 Hz/°C. Sun [20] attempted to isolate thermal stress using a thick borosilicate glass base, resulting in a temperature disturbance of 2 Hz/°C. Samarao [21] decreased the temperature disturbance from −32 ppm/°C to −10.5 ppm/°C by silicon heavy doping of at least five times, but the device layer thickness was restricted to 10 μm by complicated fabrication.

To address this problem, an all-silicon resonant pressure microsensor based on eutectic bonding was developed in this paper. More specifically, the all-silicon resonant pressure microsensor mainly consisted of an SOI wafer including a pressure-sensitive film and two resonators as well as a silicon-based vacuum cap. As the vast majority of the sensing material was silicon, the coefficient of thermal expansion was well matched, which can significantly improve the temperature performance of the resonant pressure microsensor.

## 2. Design and Simulation

The schematic of the resonant pressure microsensor is shown in Figure 1. The resonant pressure microsensor consists of an SOI wafer as the sensitive element, a silicon cap for vacuum encapsulation, and an Au layer as the bonding layer. The SOI wafer consists of a substrate layer with a pressure-sensitive film, an oxide layer with anchor points, and a device layer containing two resonators and electrodes. When pressure is applied to the pressure-sensitive film, the resonators located at the center and side of the pressure-sensitive film are subjected to tensile and compressive stresses, respectively, causing the resonant frequency of the central resonator to increase while the resonant frequency of the side resonant decreases. As part of the detection capacitor, the frequency of the resonator is converted into a current signal by a capacitor, which is driven by an electrostatic force. The drive and detection electrodes are located on both sides of the resonators, as shown in Figure 1b. As the first-order horizontal vibration mode of the resonator has a large vibration amplitude and is convenient for signal detection, the first-order mode is chosen as the operating state of the resonator.

The symmetric isolated structure on the device layer was used to match the stresses on the central and side resonators. The differential frequency shift was then used as the output to calculate the pressures to be measured.

The silicon cap for vacuum encapsulation consists of a vacuum cavity and an absorbent material for gas absorption. When the environmental temperature changes, the resonant cavity deforms in essentially the same way as the silicon-based vacuum cap, thus the resonators are not affected by thermal stresses. Therefore, the all-silicon resonant pressure microsensor is insensitive to temperature changes.

Numerical simulations of pressure and temperature characteristics of the resonant pressure microsensor were conducted using ANSYS (16.0, Ansys, Inc., Canonsburg, PA, USA). The material of the vacuum cap was determined as glass or silicon for comparison. Figure 2a shows the positive stress distribution of two resonators at a room temperature of 25 °C and under a pressure of 260 kPa. The tensile stress of the central resonator was 47.84 MPa and the compressive stress of the side resonator was 46.94 MPa. At 25 °C, when the pressure changed from 0 to 260 kPa, resonant frequencies of the central and the side resonators were shown to vary accordingly (see Figure 2b). More specifically, well matched pressure sensitivities of 50.38 Hz/kPa and −53.49 Hz/kPa were quantified for the central and side resonators, respectively. At a pressure of 260 kPa, when environmental temperatures changed from −60 °C to 80 °C, the frequencies of the central and the side resonators varied accordingly (see Figure 2c). When using glass as the packaging material, the temperature disturbances of central and side resonators were quantified as 19.46 Hz/°C and 17.85 Hz/°C, respectively. When using silicon as the packaging material, the temperature disturbances were quantified as −3.45 Hz/°C and −3.33 Hz/°C, respectively. These results indicated that the all-silicon resonant pressure microsensor could greatly reduce the side effects of temperature disturbances.

## 3. Fabrication

Figure 3a demonstrates the fabrication process of the MEMS-based resonant pressure microsensor. First, a 4-inch SOI wafer (a device layer, an oxide layer, and a substrate layer with thicknesses of 40 μm, 2 μm, and 300 μm, respectively) was cleaned using a standard wafer cleaning process to remove organic impurities and soluble ions (i). Using AZ4620 photoresist as a mask, the pressure-sensitive film was etched by DRIE on the substrate layer (ii). Then, SPR220 photoresist was used as a mask to etch the resonators on the device layer by DRIE, which were then released using gaseous HF (iii,iv). For the vacuum cap of silicon, a composite film consisting of Cr/Au (50/500 nm) was first evaporated on the silicon surface in order to assist eutectic bonding (v,vi). Then, a vacuum cavity was formed by multi-step etching and a getter layer was evaporated in the vacuum cavity to absorb various gases generated in bonding processes (vii,viii,ix). The device layer of SOI and the silicon-based vacuum cap were then eutectically bonded (x). After bonding, the electrodes were etched on the vacuum cap with DRIE (xi). Figure 3c shows the sensor chips after the above fabrication process with a side length of only 6.5 mm. A similar chip is fabricated using a vacuum cap of glass for comparison experiments.

## 4. Characterization

To verify the effects of all-silicon material on the temperature performance of the resonant pressure microsensors, two SOI wafers with the same sensing structures and fabrication steps were bonded with BF33 glass and silicon as vacuum caps for comparison. As shown in Figure 4, the bonded pressure microsensors were placed in a temperature chamber with pressures under measurements regulated by a pressure controller. A closed-loop self-oscillating circuit was used to measure the resonant frequencies of the resonators.

### 4.1. Pressure and Temperature Characterization

Figure 5a shows the resonance plots of the central and side resonators of the all-silicon resonant pressure microsensor at different temperatures. The increase in temperature leads to an increase in damping and quality factors. As the temperature increases from −40 °C to 20 and 80 °C, the quality factor of the side resonator decreases from 11,095 to 6792 and 5312, respectively, and the quality factor of the central resonator decreases from 12,195 to 7542 and 5996, respectively. Figure 5b shows the resonance plots of the central and side resonators of the all-silicon resonant pressure microsensor at different pressures. At 10 kPa, 100 kPa, and 200 kPa, the quality factors for the central resonator are 7642, 7542, and 7369, respectively, and the quality factors for the side resonator are 6971, 6792, and 6890, respectively, which means the quality factors are not significantly affected by pressure.

Figure 5c shows the frequency responses of resonant pressure microsensors relying on glass or silicon caps at a measure range of 260 kPa. At 20 °C, the pressure sensitivities of the central and side resonators with the glass cap were quantified as 58.23 Hz/kPa and −57.70 Hz/kPa, respectively, while the pressure sensitivities with the silicon cap were measured as 50.78 Hz/kPa and −50.56 Hz/kPa, respectively. The differences between the central and side resonators were mainly due to fabrication errors in DRIE. Figure 5d shows the frequency responses of resonant pressure microsensors relying on glass or silicon caps from −40 °C to 80 °C. The temperature disturbances of the resonant pressure microsensors with the glass cap were quantified as 14.47 Hz/°C of the central resonator and 19.09 Hz/°C of the side resonator, while the temperature disturbances with the silicon cap were only −0.82 Hz/°C of the central resonator and −2.36 Hz/°C of the side resonator at a pressure of 260 kPa. The temperature disturbance of the all-silicon resonant pressure microsensor was reduced by at least eight times compared with the counterparts relying on glass caps, which greatly improved the temperature performances of the resonant pressure microsensors.

### 4.2. Accuracy and Stability

The resonant pressure microsensors were calibrated within a pressure range of 0–260 kPa and a temperature range of −40 °C–80 °C, where a polynomial fitting was performed to achieve temperature compensations by double-beam differencing. As shown in Figure 6a, the maximum fitting errors of the resonant pressure microsensor based on the glass cap were quantified as 74.2 Pa, with an accuracy of 0.03% FS. In addition, the fitting errors were significantly correlated with temperature variations. For the resonant pressure microsensor with a silicon cap, the maximum fitting errors were only 11.9 Pa, with an accuracy of 0.01% FS, and the fitting errors were independent of temperature variations. Thus, the all-silicon microsensor could significantly improve measurement accuracies by eliminating the mismatches of thermal expansion coefficients.

The stability of the resonant pressure microsensors depending on the use of glass or silicon caps was examined separately over 12 h. As shown in Figure 6b, the fluctuation ranges of resonant frequencies the resonant pressure microsensors were quantified as 3.2 Hz for the glass cap and 1.5 Hz for the silicon cap. Thus, the all-silicon microsensor effectively reduced the residual thermal stress during the bonding process to improve the stability of the microsensor.

## 5. Conclusions

In this paper, a resonant pressure microsensor based on eutectic bonding of all-silicon material was designed, numerically simulated, fabricated, and experimentally characterized. The experimental characterization showed that the temperature disturbance of the all-Si material microsensor was at least eight times lower than that of the microsensor using the glass cap. The accuracy of the all-Si microsensor improved from 0.03% FS to 0.01% FS and the short-term frequency fluctuation decreased from 3.2 Hz to 1.5 Hz compared with the microsensor using the glass cap. The results confirmed that the temperature characteristics of the all-silicon material microsensor significantly improved in terms of accuracy and stability compared with the non-all-silicon counterparts.

## Figures and Tables

**Figure 1 micromachines-14-00441-f001:**
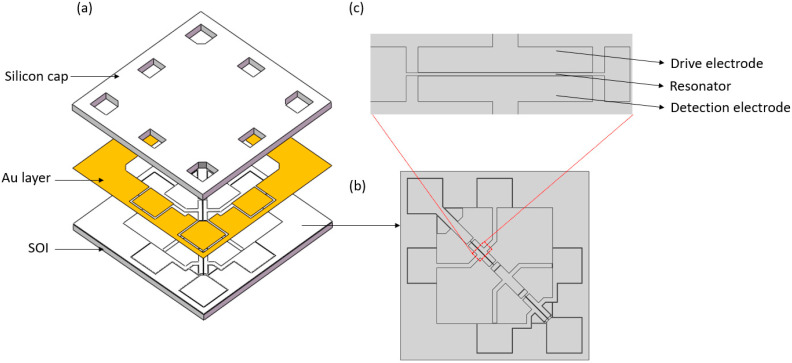
(**a**) Schematic of the resonant pressure microsensor, which consists of an SOI wafer, an Au layer, and a silicon cap. (**b**) The device layer of SOI consists of two resonators, electrodes, and stress balance structures. (**c**) The drive and detection electrodes are located on either side of the resonator.

**Figure 2 micromachines-14-00441-f002:**
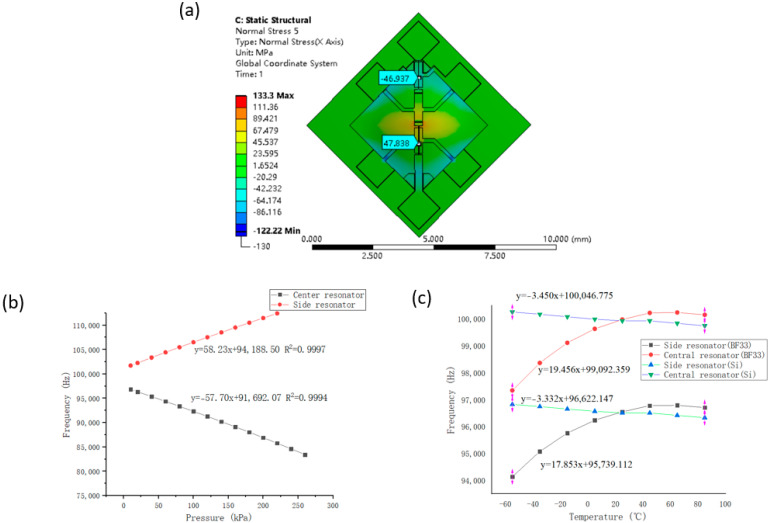
Numerical simulations: (**a**) positive stress distribution of the central and side resonators on the device layer under a pressure of 260 kPa, (**b**) resonant frequencies of the central and side resonators as a function of pressures under measurements, and (**c**) resonant frequencies of the central and side resonators with silicon or glass encapsulation as a function of temperature variations.

**Figure 3 micromachines-14-00441-f003:**
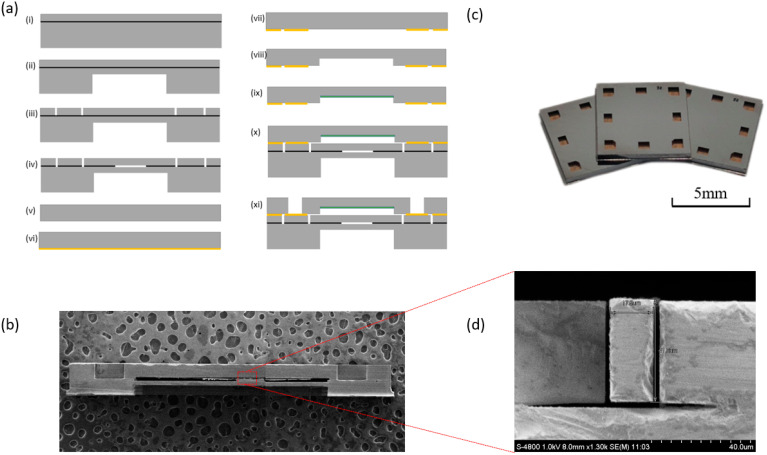
(**a**) Schematic of the fabrication flow chart: (i) cleaning SOI, (ii) etching pressure-sensitive film, (iii) etching device layer to form resonators, (iv) releasing resonators, (v) cleaning silicon wafer, (vi) evaporating Cr/Au, (vii) patterning Cr/Au, (viii) etching vacuum cavity, (ix) evaporating getter, (x) eutectic bonding, and (xi) etching electrodes. (**b**) SEM image of the cross section of the chip. (**c**) Images of fabricated sensor chips. (**d**) A cross-sectional SEM image of the resonator.

**Figure 4 micromachines-14-00441-f004:**
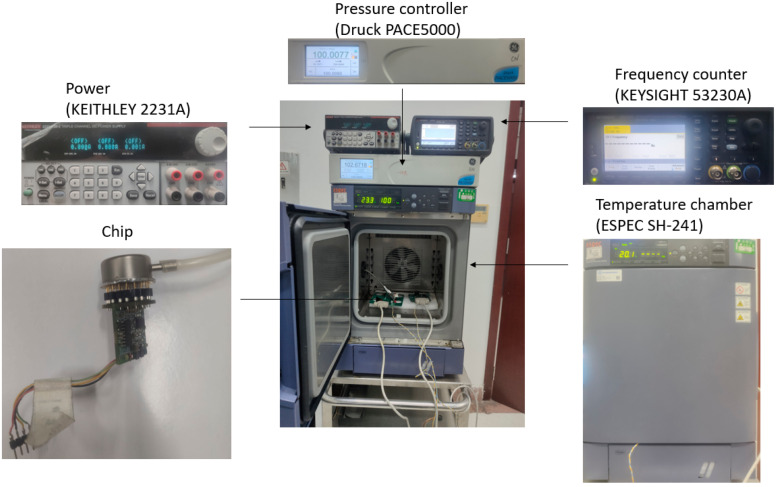
Test platform. Pressures under measurements and temperature disturbances were controlled by a pressure controller and a temperature chamber, respectively. Meanwhile, changes in resonant frequencies were detected using a closed-loop circuit.

**Figure 5 micromachines-14-00441-f005:**
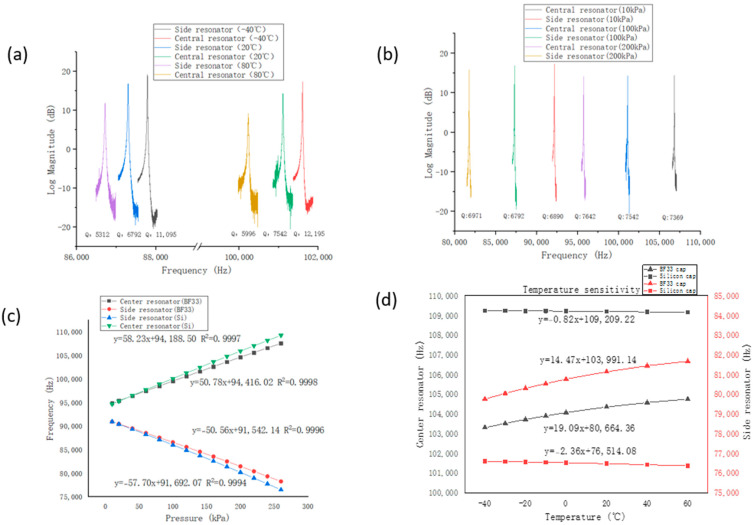
Experimental results. (**a**) Resonance plots of the central and side resonators of the all-silicon resonant pressure microsensor at different temperatures; (**b**) resonance plots of the central and side resonators of the all-silicon resonant pressure microsensor at different pressures; (**c**) resonant frequencies of the central and side resonators with silicon or glass encapsulation as a function of varied pressures (0–260 kPa) under a temperature of 20 °C; (**d**) resonant frequencies of the central and side resonators with silicon or glass encapsulation as a function of temperature variations (−40–80 °C) under a pressure of 260 kPa.

**Figure 6 micromachines-14-00441-f006:**
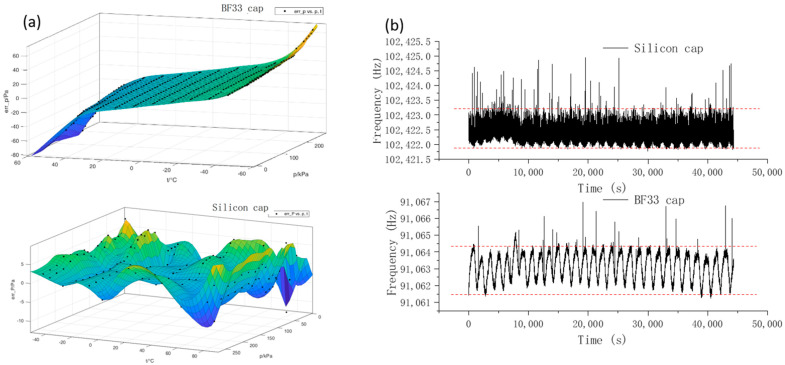
Accuracy (**a**) and stability (**b**) of the pressure-sensitive microsensors with glass or silicon caps.

## Data Availability

Not applicable.

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
