# Peer review of "An All-Silicon Resonant Pressure Microsensor Based on Eutectic Bonding"

_micromachines, 2023, doi:10.3390/mi14020441_

Round 1
Reviewer 1 Report
In this manuscript, the authors reported an all-silicon resonant pressure microsensor based on eutectic bonding, which can eliminate thermal expansion coefficient mismatches and residual thermal stresses during the bonding process. The reviewer believes that this manuscript has a merit for publication in this journal. Please address the following comments to clarify the manuscript prior to publication.
1. The authors are recommended to improve the clarity of some of the images. For example, the schematics in Figure 1 (a) and (b), numerical simulation in Figure 2(a). The current presentation of Figure 1 (a) and (b) and Figure 2(a) are not clear even for people who work in this field.
2. A cross-sectional SEM image of the sensor will be informative for the readers. This can be added to the current Figure 3.
3. The zoom-in image of the test platform in Figure 4 will also be informative for the readers.
4. The working principle of all resonators in this manuscript needs to be explained in a lot more detail along with clear illustrations. Include clear images of the resonators, explain the transduction mechanism, present the mode of vibrations, and clear SEM images of the fabricated resonators.
5. Since numbers are provided in Figure 3 (a), the authors are recommended to refer to the numbers when narrating the fabrication process flow in Section 3.
6. Basic parameters of the resonators, such as resonance plots, quality factors, and insertion losses must be presented in the manuscript.
7. The authors are recommended to also include plots that indicate the changes of quality factor and insertion loss as a function of pressure and temperature.
8. What limits the pressure measurement to 260 kPa and temperature measurement from -40 to 80 degrees Celsius?
9. How does this resonant pressure sensor behave at negative pressures?
10. What are the resolution and detection limits of this resonant pressure sensor?
11. What factors limit the resolution and detection limits of this resonant pressure sensor?
Reviewer 2 Report
This paper proposes a new design for resonant pressure microsensors to reduce the effects of temperature change on the measurement procedure. To this goal, the resonator and the cap are considered to be made of the same material (i.e. silicon). Numerical analysis is performed using ANSYS and the findings are confirmed by experiment. Results indicate that the proposed idea can reduce the measurement error caused by temperature change considerably and can improve the accuracy of the output results.
I believe that this paper presents an interesting idea and the authors could clearly describe their work and support it utilizing numerical calculations and experiment.
Hence, in my opinion, this paper can be accepted for publication.
Author Response
Thanks for your suggestions!
Round 2
Reviewer 1 Report
In this manuscript, the authors reported an all-silicon resonant pressure microsensor based on eutectic bonding, which can eliminate thermal expansion coefficient mismatches and residual thermal stresses during the bonding process. This manuscript is a revised version of the previously reviewed micromachines-2183876. The reviewer believes that sufficiently address comments provided in the first peer-review process. This manuscript can be accepted for publication.